# Fast Adaptation of Manipulator Trajectories to Task Perturbation by Differentiating through the Optimal Solution

**DOI:** 10.3390/s22082995

**Published:** 2022-04-13

**Authors:** Shashank Srikanth, Mithun Babu, Houman Masnavi, Arun Kumar Singh, Karl Kruusamäe, Krishnan Madhava Krishna

**Affiliations:** 1Robotics Research Center, KCIS, IIIT Hyderabad, Hyderabad 500032, India; s.shashank2401@gmail.com (S.S.); mithunbabu1141995@gmail.com (M.B.); mkrishna@iiit.ac.in (K.M.K.); 2Institute of Technology, University of Tartu, 50090 Tartu, Estonia; houman.masnavi@ut.ee (H.M.); karl.kruusamae@ut.ee (K.K.)

**Keywords:** manipulation, task perturbation, optimization, control

## Abstract

Joint space trajectory optimization under end-effector task constraints leads to a challenging non-convex problem. Thus, a real-time adaptation of prior computed trajectories to perturbation in task constraints often becomes intractable. Existing works use the so-called warm-starting of trajectory optimization to improve computational performance. We present a fundamentally different approach that relies on deriving analytical gradients of the optimal solution with respect to the task constraint parameters. This gradient map characterizes the direction in which the prior computed joint trajectories need to be deformed to comply with the new task constraints. Subsequently, we develop an iterative line-search algorithm for computing the scale of deformation. Our algorithm provides near real-time adaptation of joint trajectories for a diverse class of task perturbations, such as (i) changes in initial and final joint configurations of end-effector orientation-constrained trajectories and (ii) changes in end-effector goal or way-points under end-effector orientation constraints. We relate each of these examples to real-world applications ranging from learning from demonstration to obstacle avoidance. We also show that our algorithm produces trajectories with quality similar to what one would obtain by solving the trajectory optimization from scratch with warm-start initialization. Most importantly, however, our algorithm achieves a worst-case speed-up of 160x over the latter approach.

## 1. Introduction

A change in task-specification is often unavoidable in real-world manipulation problems. For example, consider a scenario where a manipulator is handing over an object to a human. The robot’s estimate of the goal position can change as it executes its prior computed trajectories. Consequently, it needs to quickly adapt its joint motions to reach the new goal position. In this paper, we model motion planning as a parametric optimization problem wherein the task specifications are encoded in the parameters. In this context, adaptation to a new task requires re-computing the optimal joint trajectories for the new set of parameters. This is a computationally challenging process as the underlying cost functions in typical manipulation tasks are highly non-linear and non-convex [1]. Existing works leverage the so-called warm-starting technique where prior computed trajectories are used as initialization for the optimization solvers [2]. However, our extensive experimentation with off-the-shelf optimization solvers such as Scipy-SLSQP [3] show it is not sufficient for real-time adaptation of joint trajectories to task perturbations.

### 1.1. Main Idea

The proposed work explores an alternate approach based on differentiating the optimal solution with respect to the problem parameters, hereafter referred to as the Argmin differentiation [4]. To understand this further, consider the following constrained optimization problem over variable ξ (e.g., joint angles) and parameter vector p (e.g., end-effector position).
(1)$$\begin{array}{c} \xi^{*}\left(p\right)=\arg\min f\left(\xi ,p\right)\; \end{array}$$
(2)$$\begin{array}{c} g_{i}\left(\xi ,p\right)\leq 0,\forall i=1,2,\cdots n \end{array}$$
(3)$$\begin{array}{c} h_{j}\left(\xi ,p\right)=0,\forall j=1,2,\cdots m \end{array}$$

The optimal solution $\xi^{*}$ satisfies the following Karush–Kuhn Tucker (KKT) conditions.
(4a)$$\begin{array}{c} \nabla f\left(\xi^{*},p\right)+\sum_{i}\lambda_{i}\nabla g_{i}\left(\xi^{*},p\right)+\sum_{j}\mu_{j}\nabla h_{j}\left(\xi^{*},p\right)=0 \end{array}$$
(4b)$$\begin{array}{c} g_{i}\left(\xi^{*},p\right)\leq 0,\forall i \end{array}$$
(4c)$$\begin{array}{c} h_{j}\left(\xi^{*},p\right)=0 \end{array}$$
(4d)$$\begin{array}{c} \lambda_{i}\geq 0,\lambda_{i}g_{i}\left(\xi^{*},p\right)=0,\forall i. \end{array}$$

The gradients in ([Disp-formula FD4a-sensors-22-02995]) are taken with respect to ξ. The variables $\lambda_{i},\mu_{j}$ are called the Lagrange multipliers. Now, consider a scenario where the optimal solution $\xi^{*}$ for the parameter p needs to be adapted for the perturbed set $\bar{p}=p+\Delta p$. As mentioned earlier, one possible approach is to resolve the optimization with $\xi^{*}$ as the warm-start initialization. Alternately, for Δp with a small magnitude, an analytical perturbation model can be constructed. To be precise, we can compute the first-order differential of the r.h.s. of ([Disp-formula FD4a-sensors-22-02995])–([Disp-formula FD4d-sensors-22-02995]) to obtain analytical gradients in the following form [5,6,7].
(5)$$\begin{array}{c} \left(\nabla_{p}\xi^{*},\nabla_{p}\lambda_{i},\nabla_{p}\mu_{i}^{*}\right)=F\left(\xi^{*},p,\lambda_{i},\mu_{j}\right) \end{array}$$

Multiplying the gradients with Δp gives us an analytical expression for the new solution and Lagrange multipliers corresponding to the perturbed parameter set [7].

### 1.2. Contribution

**Algorithmic Contribution:** A critical bottleneck in using the gradient map of the form (Equation 5) to compute perturbed solutions is that the mapping between Δp and $\lambda_{i}$ is highly discontinuous. In other words, even a small Δp can lead to large changes in the so-called active-set of the inequality constraints. Thus it becomes necessary to develop additional active-set prediction mechanisms [7]. In this paper, we bypass this complication by instead focusing on the parametric optimization with only bound constraints on the variable set. Argmin differentiation of such problems has a simpler structure, which we leverage to develop a line-search based algorithm to incrementally adopt joint trajectories to larger changes in the parameter/tasks. To give some example of “large perturbation”, our algorithm can adapt the joint trajectories of Franka–Panda arm to a perturbation of up to 30 cm in the goal position. This is almost 30% of the workspace of the Franka arm.

**Application Contribution**: For the first time, we apply the Argmin differentiation concept to the problem of joint trajectory optimization for the manipulators under end-effector task constraints. We consider a diverse class of cost functions to handle (i) perturbations in joint configurations or (ii) end-effector way-points in orientation-constrained end-effector trajectories. We present an extensive benchmarking of our algorithm’s performance as a function of the perturbation magnitude. We also show that our algorithm outperforms the warm-start trajectory optimization approach in computation time by several orders of magnitude while achieving similar quality as that measured by task residuals and smoothness of the resulting trajectory.

### 1.3. Related Works

The concept of Argmin differentiation has been around for a few decades, although often under the name of sensitivity analysis [8,9]. However, of late it has seen a resurgence, especially in the context of end-to-end learning of control policies [10,11]. Our proposed work is more closely related to those that use Argmin differentiation for motion planning or feedback control. In this context, a natural application of Argmin differentiation is in bi-level trajectory optimization where the gradients of the optimal solution from the lower level are propagated to optimize the cost function at the higher level. This technique has been applied to both manipulation and navigation problems in existing works [6,12]. Alternately, Argmin differentiation can also be used for the correction of prior-computed trajectories [7,13].

To the best of our knowledge, we are not aware of any work that uses Argmin differentiation for the adaptation of task-constrained manipulator joint trajectories. The closest to our approach is [5] that uses it to accelerate the inverse kinematics problem. Along similar lines, [7] considers a very specific example of perturbation in the end-effector goal position. In contrast to these two cited works, we consider a much more diverse class of task constraints. Furthermore, our formulation also has important distinctions with [7] at the algorithmic level. Authors in [7] use the log-barrier function for including inequality constraints as penalties in the cost function. In contrast, we note that in the context of the task-constrained trajectory optimization considered in this paper, the joint angle limits are the most critical. The velocity and acceleration constraints can always be satisfied through time-scaling based pre-processing [14]. Thus, by choosing a way-point parametrization for the joint trajectories, we formulate the underlying optimization with just box constraints on the joint angles. This, in turn, allows us to treat this constraint through simple projection (Line 4 in Algorithm 1) without disturbing the structure of the cost function and the resulting Jacobian and Hessian matrices obtained by Argmin differentiation.
**Algorithm 1** Line-Search Based Joint Trajectory Adaptation to Task Perturbation1:Initialize ${}^{k}\xi^{*}$ as the solution for the prior parameter ${}^{k}p$, the Hessian ${}^{k}\nabla_{\xi }^{2}f\left({}^{k}\xi ,p\right)$, the gradient $\nabla_{\xi ,p_{i}}f\left({}^{k}\xi ,p\right)$, and ${}^{k}\Delta p=\bar{p}-{}^{k}p$2:**while**η>0**do**(6a)$$\begin{array}{c} \max\eta \end{array}$$(6b)$$\begin{array}{c} f({}^{k}\xi^{*}\left(p+\eta \Delta p\right),p+\Delta p)\leq f\left({}^{k}\xi^{*},p+\Delta p\right) \end{array}$$3:    
(7)$$\begin{array}{c} {}^{k+1}\xi^{*}={}^{k}\xi^{*}+\eta \nabla_{p}\xi^{*}\Delta {}^{k}p \end{array}$$4:    
(8)$$\begin{array}{c} {}^{k+1}\xi^{*}=Project\left(\xi_{lb},\xi_{ub}\right) \end{array}$$5:    Update ${}^{k+1}\bar{p}$ = $ForwardRoll({}^{k+1}\xi^{*})$6:    Update ${}^{k+1}\Delta p=\bar{p}-{}^{k+1}p$.7:    Update Hessian $\nabla_{\xi }^{2}f\left({}^{k+1}\xi ,{}^{k+1}p\right)$.8:    Update Jacobian $\nabla_{\xi ,p}f\left({}^{k+1}\xi ,{}^{k+1}p\right)$9:**end while**

## 2. Proposed Approach

### 2.1. Symbols and Notations

We will use lower case normal font letters to represent scalars, while bold font variants will represent vectors. Matrices are represented by upper case bold fonts. The subscript *t* will be used to denote the time stamp of variables and vectors. The superscript *T* will represent the transposing of a matrix.

### 2.2. Argmin Differentiation for Unconstrained Parametric Optimization

We consider the optimal joint trajectories to be the solution of the following bound-constrained optimization with parameter p.
(9a)$$\begin{array}{c} \xi^{*}\left(p\right)=\arg\min_{\xi }f\left(\xi ,p\right) \end{array}$$
(9b)$$\begin{array}{c} \xi_{lb}\leq \xi \leq \xi_{ub} \end{array}$$

We are interested in computing the Jacobian of $\xi^{*}\left(p\right)$ with respect to p. If we ignore the bound-constraints for now, we can follow the approach presented in [4] to obtain them in the following form.
(10)$$\begin{array}{c} \nabla_{p}\xi =-{\left(\nabla_{\xi }^{2}f\left(\xi ,p\right)\right)}^{-1}\left[\begin{array}{ccc} \nabla_{\xi ,p_{1}}f\left(\xi ,p\right), & \cdots & \nabla_{\xi ,p_{n}}f\left(\xi ,p\right) \end{array}\right] \end{array}$$

Using (Equation 10), we can derive a local model for the optimal solution corresponding to a perturbation Δp as
(11)$$\begin{array}{c} \xi^{*}\left(\bar{p}\right)=\xi^{*}\left(p\right)+\nabla_{p}\xi^{*}\overset{\Delta p}{\overset{︷}{\left(\bar{p}-p\right)}}, \end{array}$$

Intuitively, (Equation 11) signifies a step of length Δp along the gradient direction. However, for (Equation 11) to be valid, the step-length needs to be small. In other words, the perturbed parameter $\bar{p}$ needs to be in the vicinity of p. Although it is difficult to mathematically characterize the notion of “small”, in the following, we attempt a practical definition based on the notion of optimal cost.

**Definition** **1.**
*A valid |Δp| is one that satisfies the following relationship*

(12)
$$\begin{array}{c} f\left(\xi^{*}\left(\bar{p}=p+\Delta p\right),p+\Delta p\right)\leq f\left(\xi^{*},p+\Delta p\right) \end{array}$$



The underlying intuition in (Equation 12) is that the perturbed solution should lead to a lower cost for the parameter p+Δp as compared to $\xi^{*}$ for the same perturbed parameter.

### 2.3. Line Search and Incremental Adaption

Algorithm 1 couples the concept from the definition (Equation 11) with a basic line-search to incrementally adapt (Equation 11) to a large Δp. The algorithm begins by initializing the optimal solution ${}^{k}\xi$ and the parameter ${}^{k}p$ with prior values for iteration k=0. These variables are then used to initialize the Hessian and Jacobian matrices. The core computations takes place in line 2, wherein we compute the least amount of scaling that needs to be done to step length ${}^{k}\Delta p={}^{k}\bar{p}-p$ to guarantee a reduction in the cost. At line 3, we update the optimal solution based on step-length $\eta {}^{k}\Delta p$ obtained in line 2, followed by a simple projection at line 4 to satisfy the minimum and maximum bounds. At line 5, we perform the called forward roll-out of the solution to update the parameter set. For example, if the parameter p models position of the end-effector at the final time instant of a trajectory, then line 5 computes how close the ${}^{k+1}\xi^{*}$ takes the end-effector to the perturbed goal position $\bar{p}$. On lines 7 and 8, we update the Hessian and the Jacobian matrices based on the updated parameter set and optimal solution.

## 3. Task Constrained Joint Trajectory Optimization

This section formulates various examples of the task-constrained trajectory optimization problem and uses the previous section’s results for optimal adaptation of joint trajectories under task perturbation. To formulate the underlying costs, we adopt the way-point parametrization and represent the joint angles at time *t* as $q_{t}$. Furthermore, we will use $\left(x_{e}\left(q_{t}\right),o_{e}\left(q_{t}\right)\right)$ to describe the end-effector position and orientation in terms of Euler angles, respectively.

### 3.1. Orientation Constrained Interpolation between Joint Configurations

The task here is to compute an interpolation trajectory between a given initial $q_{0}$ and a final joint configuration $q_{m}$ while maintaining a specified orientation $o_{d}$ for the end-effector at all times. We model it through the following cost function.
(13)$$\begin{array}{c} \sum_{t}f_{s}\left(q_{t-k:t}\right)+{∥\begin{array}{c} q_{t_{1}}-q_{0}\\ q_{t_{m}}-q_{m} \end{array}∥}_{2}^{2}+\sum_{t}{∥o_{e}\left(q_{t}\right)-o_{d}∥}_{2}^{2} \end{array}$$

The first term the cost function models smoothness in terms of joint angles from t−k to *t* [15]. For example, for k=1, the smoothness is defined as the first-order finite difference of the joint positions at subsequent time instants. Similarly, k=2,3, will model higher order smoothness through second and third-order finite differences respectively. We consider all three finite-differences in our smoothness cost term. The second term ensures that the interpolation trajectory is close to the given initial and final points. The final term in the cost function maintains the required orientation of the end-effector.

We can shape (Equation 13) in the form of ([Disp-formula FD9a-sensors-22-02995]) by defining $\xi =(q_{t_{1}},q_{t_{2}},\cdots q_{t_{m}})$. The bounds will correspond to the maximum and minimum limits on the joint angles at each time instant. We define the parameter set as $p=(q_{0},q_{m})$. That is, we are interested in computing the adaptation when either or both of $q_{0}$ and $q_{m}$ gets perturbed.

#### Applications

Adaptation of $\xi^{*}$ of (Equation 13) for different $q_{0},q_{m}$ has applications in learning from demonstration setting where the human just provides the information about the initial and/or final joint configuration, and the manipulator then computes a smooth interpolation trajectory between the boundary configurations by adapting a prior computed trajectory.

Figure 1 presents an example of adaptation discussed above. The prior computed trajectory is shown in blue. This is then adapted to two different final joint configurations. The trajectory computed through Algorithm 1 is shown in green, while that obtained by resolving the optimization problem (with warm-starting) is shown in red.

### 3.2. Orientation-Constrained Trajectories through Way-Points

The task in this example is to make the end-effector move though given way-points while maintaining the orientation at $o_{d}$. Let $x_{d_{t}}$ represent the desired way-point of the end-effector at time *t*. Thus, we can formulate the following cost function for the current task.
(14)$$\begin{array}{c} \sum_{t}f_{s}\left(q_{t-k:t}\right)+\sum_{t}∥o_{e}\left(q_{t}\right)-o_{d}{∥}_{2}^{2}+\sum_{t}{∥x_{e}\left(q_{t}\right)-x_{d_{t}}∥}_{2}^{2} \end{array}$$

The first two terms in the cost function are the same as the previous example. The changes appear in the final term which minimizes the $l_{2}$ norm of the distance of the end-effector with the desired way-point. The defintion of ξ remains the same as before. However, the parameter set is now defined as $p=(x_{d_{1}},x_{d_{2}},\cdots x_{d_{m}})$.

#### Application

**Collision Avoidance** As shown in Figure 2, a key application of the adaptation problem discussed above is in collision avoidance. A reactive planner such as [16] can provide new via-points for the manipulator to avoid collision. Our Algorithm 1 can then use the cost function (Equation 14) to adapt the prior trajectory shown in blue to that shown in green. For comparison, the trajectory obtained with resolve of the trajectory optimization is shown in red.

**Human–Robot Handover:** Algorithm 1 with cost function (Equation 14) also finds application in human–robot handover tasks. An example is shown in Figure 3, where the manipulator adapts the prior trajectory (blue) to a new estimate of the handover position. As before, the trajectory obtained with Algorithm 1 is shown in green, while the one shown in red corresponds to a re-solve of the trajectory optimization with warm-start initialization.

## 4. Benchmarking

### 4.1. Implementation Details

The objective of this section is to compare the trajectories computed by Algorithm 1 with that obtained by re-solving the trajectory optimization for the perturbed parameters with warm-start initialization. We consider the same three benchmarks presented in Figure 1, Figure 2 and Figure 3 implemented on a 7dof Franka Panda Arm, but for a diverse range of perturbations magnitude. For each benchmark, we created a data set of 180 trajectories by generating random perturbations in the task parameters. For the benchmark of Figure 1, the parameters are the joint angles, but in the following we use the forward kinematics to derive equivalent representation for the parameters in terms of end-effector position values.

Each joint trajectory is parameterized by a 50-dimensional vector of way-points. Thus, the underlying task constrained trajectory optimization involves a total of 350 variables. We use Scipy-SLSQP [3] to obtain the prior trajectory and also to re-solve the trajectory optimization for the perturbed parameters. We did our implementation in Python using Jax-Numpy [17] to compute the necessary Jacobian and Hessian matrices. We also used the just-in-time compilation ability of JAX to create an on-the-fly compiled version of our codes. The line-search in Algorithm 1 (line 2) was done through a parallelized search over a set of discretized η values. The entire implementation was done on a 32 GB RAM i7-8750 desktop with RTX 2080 GPU (8GB). To foster further research in this field and ensure reproducibility, we open-source our implementation for review at https://rebrand.ly/argmin-planner (First released on 15 September 2020).

### 4.2. Quantitative Results

**Orientation Metric:** For this analysis, we compared the pitch and roll angles at each time instant along trajectories obtained with Algorithm 1 and the resolving approach. Specifically, we computed the maximum of the absolute difference (or $L_{\infty }$ norm) of the two orientation trajectories. The yaw orientation in all these benchmarks was a free variable and is thus not included in the analysis. The results are summarized in Figure 4, Figure 5 and Figure 6. The histogram plot in these figures are generated for the medium perturbation ranges (note the figure legends). For the Figure 1 benchmark related to cost function (Equation 13), Figure 4 shows that all the trajectories obtained by Algorithm 1 have $L_{\infty }$ norm of the orientation difference less than 0.1 rad. For the benchmark of Figure 2, which we recall involves perturbing the via-point of the end-effector trajectory, the histograms of Figure 5 show similar trends. All the trajectories computed by Algorithm 1 managed a similar orientation difference. For the benchmark of Figure 3 pertaining to the perturbation of the final position, 69.41% of the trajectories obtained by Algorithm 1 managed to maintain a orientation difference of 0.1 rad with the resolving approach.

**Task residuals ratio metric:** For this analysis, we compare the task residual between trajectories obtained from Algorithm 1 and the resolving approach. For example, for the benchmark of Figure 1, we want the manipulator final configuration to be close to the specified value (recall cost (Equation 13)) while maintaining the desired orientation at each time instant. Thus, we compute the $L_{\infty }$ residual of $q_{t}-q_{m}$ for Algorithm 1 and compare it with that obtained from the resolving approach. Now, as previously, and to be consistent with the other benchmarks, we convert the residual of the joint angles to position values through forward kinematics. Similar analysis follow for the other benchmarks as well. For the ease of exposition, we divide the task residual of Algorithm 1 by that obtained with the resolving approach. A ratio greater than 1 implies that the former led to a higher task residual than the latter and vice-versa. Similarly, a ratio closer to 1 implies that both the approaches performed equally well.

The results are again summarized in Figure 4, Figure 5 and Figure 6. From Figure 4, we notice that 97.05% of trajectories have a residual ratio less than 1.2. For the experiment involving via-point perturbation in Figure 5, the performance drops to 62.50% for the same value of residual ratio. Meanwhile, as shown in Figure 6, around 82.94% of the trajectories have a residual ratio less than 1.2 in the case of the final position perturbation benchmark of Figure 3.

**Velocity Smoothness Metric:** For this analysis, we computed the difference in the velocity smoothness cost ($L_{2}$ norm of first-order finite difference) between the trajectories obtained with Algorithm 1 and the resolving approach. The results are again summarized in Figure 4, Figure 5 and Figure 6. For all the benchmarks, in around 65% of the examples, the difference was less than 0.05. This is 35% of the average smoothness cost observed across all the trajectories from both the approaches.

**Scaling with Perturbation Magnitude:** The line plots in Figure 4, Figure 5 and Figure 6 represent the first quartile, median and the third quartile of the three metrics discussed above for different perturbation ranges.

For the benchmark of Figure 1, trajectories from Algorithm 1 maintains an orientation difference of less than 0.1 rad, with the trajectories of the resolving approach for perturbations as large as 40 cm. The difference in smoothness cost for the same range is also small, with the median value being in the order of $10^{-3}$. The median task residuals achieved by Algorithm 1 is only 2% higher than that obtained by the resolving approach. For the benchmark of Figure 2, the performance remains same on the orientation metric, but the median difference in smoothness cost and task residual ration increases to 0.04 and 9% for the largest perturbation range. The benchmark of Figure 6 follows a similar trend in orientation and smoothness metric, but performs significantly worse in task residuals. For the largest perturbation range, Algorithm 1 leads to 50% higher median task residuals. However, importantly, for perturbation up to 30 cm, the task residual ratio is close to 1, suggesting that Algorithm 1 performed as well as the resolving approach for these perturbations.

**Computation Time:**Table 1 contrasts the average timing of our Algorithm 1 with the approach of resolving the trajectory optimization with warm-start initialization. As can be seen, our Argmin differentiation based approach provides a worst-case speed up of 160x on the benchmark of Figure 3. For the rest of the benchmarks, this number varies between 500 to 1000. We believe that this massive gain in computation time offsets whatever little performance degradation in terms of orientation, smoothness, and task residual metric that Algorithm 1 incurs compared to re-solving the problem using warm-start. Note that the high computation time of the re-solving approach is expected, given that we are solving a difficult non-convex function over a long horizon of 50 steps resulting in 350 decision variables. Even highly optimized planners like [1] show similar timings on closely related benchmarks [18].

## 5. Conclusions and Future Work

We presented a fast, near real-time algorithm for adapting joint trajectories to task perturbation as high as 40 cm in the end-effector position, almost half the radius of the Franka Panda arm’s horizontal workspace used in our experiments. By consistently producing trajectories similar to those obtained by resolving the trajectory optimization problem but in a small fraction of a time, our Algorithm 1 opens up exciting possibilities for reactive motion control of manipulators in applications like human–robot handover.

Our algorithm is easily extendable to other kind of manipulators. The only requirement is that we should know the forward kinematics of the manipulator. This would allow us to get the algebraic expressions for functions $o_{e}\left(q\right)$ and $x_{e}\left(q\right)$ in cost function (13) and (14), respectively. In our implementation, we derived the forward kinematics and $o_{e}\left(q\right)$ and $x_{e}\left(q\right)$ through the DH representation of the manipulator. The DH table is available for many commercial manipulators, e.g., UR5e besides the Franka Panda Arm used in our simulation.

Our algorithm does not depend on any specific sensing modality. For example, in collision avoidance applications, we assume that obstacle information is used by some higher level planners that provides intermediate collision-free points to the manipulator, which then uses the ArgMin differentiation to replan its prior trajectories.

There are several ways to improve our algorithm. First, the joint bounds can also be included as penalties in the cost function itself, in addition to being handled by projection (Line 11 in Algorithm 1). This would ensure that the gradient and Hessian of the optimal cost is aware of the joint limit bounds. Second, we can consider a low dimensional polynomial representation of the trajectories. For example, the joint trajectories can be represented by a 10th order Bernstein polynomial with the coefficients acting as the variables of the optimization problem. This would drastically reduce the computation cost of obtaining the Hessian of the optimal cost as compared to current way-point paramaterization of the joint trajectory that requires around 50 variables to represent one joint trajectory.

In future works, we will extend our formulation to problems with dynamic constraints, such as torque bounds. We conjecture that by coupling the way-point parametrization with a multiple-shooting like approach, we can retain the constraints as simple box-bounds on the decision variables and consequently retain the computational structure of the Algorithm 1. We are also currently evaluating our algorithm’s performance on applications such as autonomous driving.

## Figures and Tables

**Figure 1 sensors-22-02995-f001:**
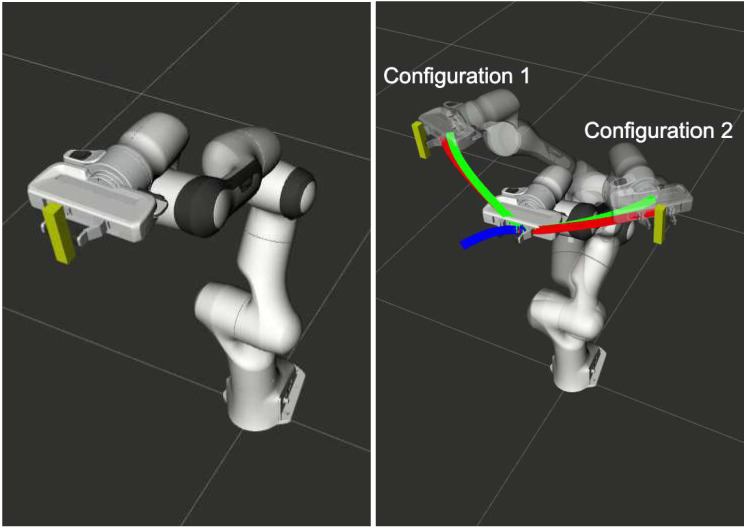
Prior trajectory shown in blue is used to adapt the joint motions to move towards two different final joint configurations while maintaining the horizontal orientation of the end-effector at all times.

**Figure 2 sensors-22-02995-f002:**
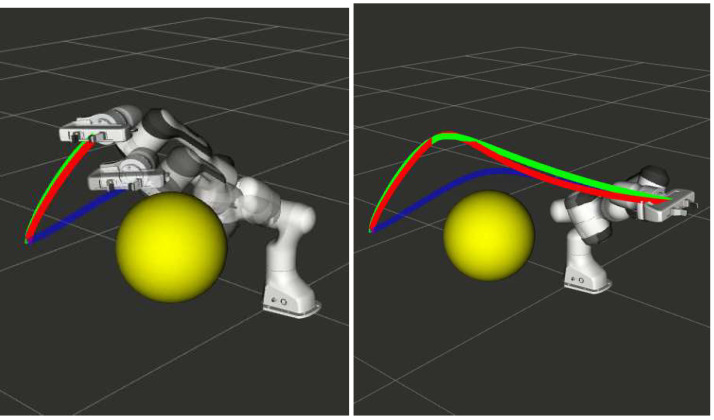
Collision avoidance by perturbing the mid-point of the prior computed end-effector trajectory.

**Figure 3 sensors-22-02995-f003:**
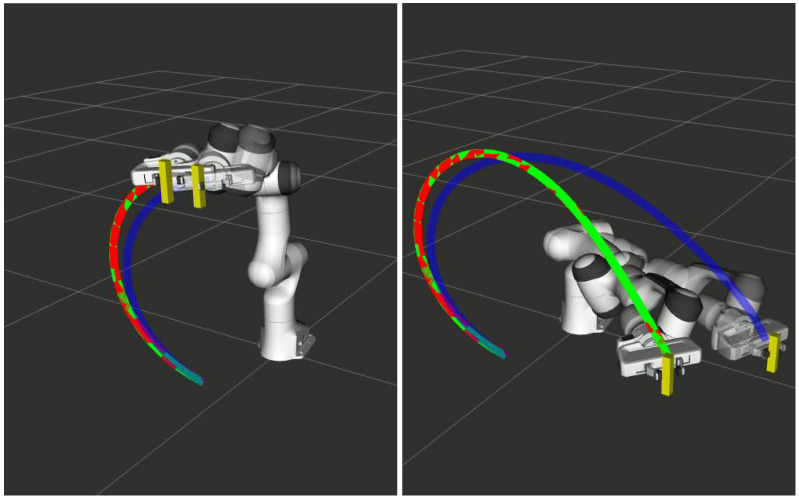
Perturbation in the final position of the end-effector.

**Figure 4 sensors-22-02995-f004:**
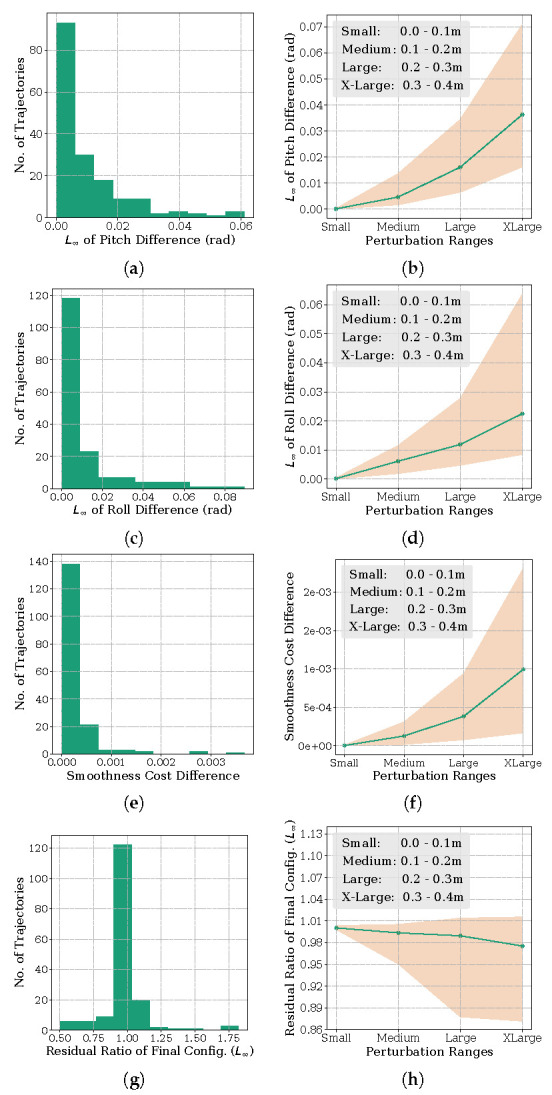
Performance of Algorithm 1 for different perturbation ranges on the benchmark of Figure 1 that involves perturbing the final joint configuration (recall cost function (Equation 13)). Note that the perturbation in the final joint is converted to position values by forward kinematics. The (**a**,**c**,**e**,**g**) column shows the histogram of orientation, smoothness and task residual ratio metrics for the medium range perturbation. The (**b**,**d**,**f**,**h**) column quantifies the metrics for different perturbation ranges.

**Figure 5 sensors-22-02995-f005:**
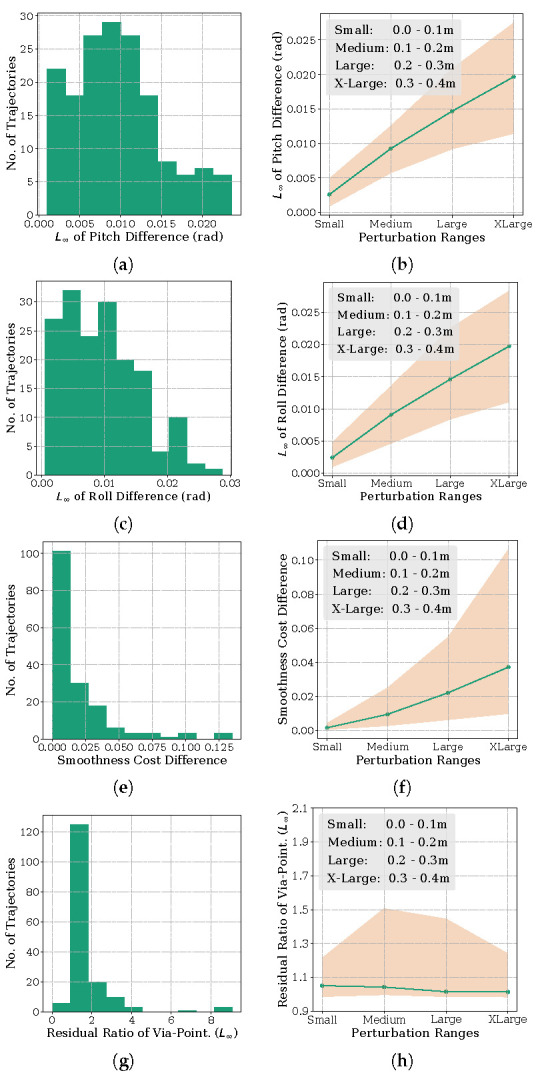
Performance of Algorithm 1 for different perturbation ranges on the benchmark of Figure 2 that involves perturbing the via-point of the end-effector trajectory (recall cost function (Equation 14)). The (**a**,**c**,**e**,**g**) and (**b**,**d**,**f**,**h**) columns show similar benchmarking as those of Figure 4.

**Figure 6 sensors-22-02995-f006:**
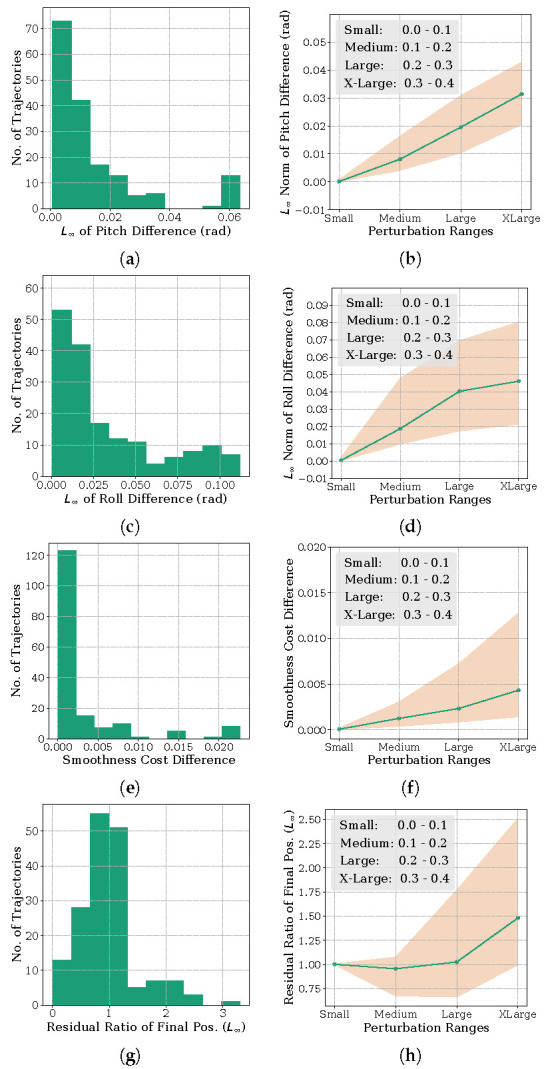
Performance of Algorithm 1 for different perturbation ranges on the benchmark of Figure 6 that involves perturbing the final end-effector position. (recall cost function (Equation 14)). The (**a**,**c**,**e**,**g**) and (**b**,**d**,**f**,**h**) columns show similar benchmarking as those of Figure 4.

**Table 1 sensors-22-02995-t001:** Computation times comparison between Algorithm 1 and resolving trajectory optimization approach on three benchmarks.

	SciPy-SLSQP		Our Algorithm 1
Benchmarks	Wall Time (s)	Wall Time w/o Jacobian and Function Evaluation Overhead (s)	Wall Time (s)
Final Configuration Perturbation (Figure 1)	43.91	41.09	**0.039**
Via Point Perturbation (Figure 2)	53.05	34.74	**0.09**
Final Position Perturbation (Figure 3)	35.91	29.09	**0.18**

## Data Availability

Codes to reproduce the results are available at https://rebrand.ly/argmin-planner (accessed on 1 August 2020).

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
