# Peer review of "Fast Adaptation of Manipulator Trajectories to Task Perturbation by Differentiating through the Optimal Solution"

_sensors, 2022, doi:10.3390/s22082995_

Round 1

Reviewer 1 Report

The authors has applied Argmin differentiation for robotic arm motion planning in collision avoidance task, which is a fundamentally different approach for manipulator trajectories optimization, based on deriving analytical gradients of the optimal solution with respect to the task constraint parameters. This algorithm provides drastic speed-up of 160x, although precision of positioning can be partially reduced and applied method uses calculation model, operating with  350 decision variables.

              The proposed method seems to be interesting enough for publication in “Sensors” mdpi journal.

Some drawback of the manuscript is that given data are disclosed in detailes for 7 dof Franka Panda Arm, but there are no any discussion, disclosing what precision result can be extrapolated for a benchmark with other sensors types, alternative number and types of  arm elements. Here, given general expressions and description of algorithm does not give obvious enough opportunity for a reader, working in distant fields. May be, the authors will manage to add such commentary.

Author Response

We are attaching a pdf file with response to reviewer 1 and summary of the associated changes made in the manuscript

Reviewer 2 Report

How to improve the convergence of the proposed decendent algoritm

Author Response

We are attaching a pdf with the response to the reviewer's comments and the summary of associated changes in the manuscript.
